# Inflammatory Profile Assessment in a Highly Selected Athyreotic Population Undergoing Controlled and Standardized Hypothyroidism

**DOI:** 10.3390/biomedicines12010239

**Published:** 2024-01-22

**Authors:** Tommaso Piticchio, Francesco Savarino, Salvatore Volpe, Antonio Prinzi, Gabriele Costanzo, Elena Gamarra, Francesco Frasca, Pierpaolo Trimboli

**Affiliations:** 1Endocrinology Section, Department of Clinical and Experimental Medicine, Garibaldi Nesima Hospital, University of Catania, 95124 Catania, Italy; francescosavarino1997@gmail.com (F.S.); salvatore.volpe96@hotmail.it (S.V.); a.prinzi95@gmail.com (A.P.); dott.gabrielecostanzo@gmail.com (G.C.); f.frasca@unict.it (F.F.); 2Servizio di Endocrinologia e Diabetologia, Ospedale Regionale di Lugano, Ente Ospedaliero Cantonale (EOC), 6900 Lugano, Switzerland; elena.gamarra@eoc.ch (E.G.); pierpaolo.trimboli@eoc.ch (P.T.); 3Facoltà di Scienze Biomediche, Università della Svizzera Italiana (USI), 6900 Lugano, Switzerland

**Keywords:** inflammation, hypothyroidism, inflammatory index, lymphocytes, neutrophils, platelets, monocytes

## Abstract

**Background:** Hypothyroidism (hT) presents heterogeneous symptoms and findings. Evidence on this topic comes mainly from heterogeneous populations in terms of disease duration, residual thyroid function, and comorbidities. Therefore, it would be useful to assess systemic inflammation in a homogeneous hT population. The aim of this study was to investigate inflammation in a population that underwent standardized controlled hT. **Methods:** We recruited thyroidectomized patients diagnosed with thyroid cancer who were otherwise fit and healthy, showing hypothyroidism before I131 treatment using a standard protocol of LT4 withdrawal. The blood inflammatory indexes (BIIXs) (i.e., NLR, PLR, MLR, SII, SIRI, and AISI) were calculated using the blood tests collected just before I131 administration. Patients were divided according to sex, BMI, and thyroglobulin. The relationships between the BIIXs, age, and thyroid hormones were also investigated. **Results:** We included 143 patients. The median age of the sample was 43 years. The BIIX median values showed significant differences based on sex, BMI, and thyroglobulin levels (*p* < 0.05). No significant correlations were found between the BIIXs and age, TSH, FT4, and FT3. **Conclusions:** This study shows the BIIX median values of a population which underwent standardized hT. It suggests a role for some BIIXs in the evaluation of hypothyroidism in obese people and as hypothetical prognostic markers for thyroid cancer.

## 1. Introduction

Hypothyroidism (hT) is a disease including heterogeneous clinical symptoms and findings. According to the biochemical pattern, it is classically classified into clinical (i.e., a combination of a low serum FT4 and an elevated serum TSH) and subclinical (i.e., the serum FT4 is normal or low-normal, and the serum TSH is slightly elevated) hT [1]. Clinical or overt hT in adulthood could result in the following widespread organ-specific effects: diminished calorigenesis and oxygen consumption; impaired cardiac, pulmonary, renal, gastrointestinal, and neurological functions; and pathologicaldeposition of glycosaminoglycans in intracellular spaces, particularly in skin and muscle [2]. The signs and symptoms are nonspecific and can vary in individual presentations, ranging from states of fatigue to somnolence, arthralgias, psychasthenia, cognitive and memory impairment, cold intolerance, constipation, dry skin, hair changes (i.e., dryness, thinning, or loss), loss of libido, menorrhagia, muscle cramps, myalgia, voice changes, malnutrition, obesity, anemia, hypertension, oedema, chronic cardiac disease, and finally myxoedema coma [3,4]. Subclinical hT, on the other hand, is usually characterized only by biochemical findings, and for years, the literature has debated the possibility of finding mild-to-moderate symptomatology in selected patients [5]. Thus, clinical pictures result from disease factors (i.e., severity, duration, and rapidity of onset of thyroid hormone deficiency) and the subject’s factors (e.g., age of patient, comorbidity, lifestyle, constitutional predisposition, culture, and sensibility to pathology). This clinical landscape is further complicated by the increasing frequent onset of central or primary hT in patients treated with new anti-neoplastic drugs [6,7,8]. Therefore, evidence on the clinical presentation of hT is derived mostly from heterogeneous populations, especially in terms of duration of the disease and the residual percentage of glandular parenchyma still functioning (e.g., patients with thyroiditis) [9]. In this complex context, it would be useful to study indices that can objectify systemic inflammation in patients with controlled and standardized clinical hT.

Recently, novel blood inflammatory indexes (hereafter BIIXs) based on the correlation among blood count parameters, such as the neutrophil-to-lymphocyte ratio (NLR), platelet-to-lymphocyte ratio (PLR), monocyte-to-lymphocyte ratio (MLR), systemic immune inflammatory index (SII), system inflammation response index (SIRI), and aggregate index of systemic inflammation (AISI), have been investigated in almost all fields of clinical research and reported as reliable in quantifying inflammation, the status of systemic illness, and the prognosis [10,11,12,13,14,15,16,17,18]. Hypothyroidism is a well-known inflammatory and oxidative trigger. Since thyroid hormones have a crucial role in the metabolism and proliferation of blood cells, hT also correlates with several blood cell alterations such as anemia, erythrocytosis, leukopenia, and thrombocytopenia [19,20,21,22]. Thyroid studies on BIIXs have been mostly focused on Hashimoto thyroiditis or simple TSH variation, regardless of clinical features (i.e., patient’s clinical context, symptoms, and duration and severity of hT) [23,24,25]. Furthermore, more recent studies have been carried out in specific the setting of frail hT subjects with inflammatory responses hidden by confounding comorbidities, such as intensive care, geriatric, and oncologic cases [26,27,28]. Actually, the relationship between hT and BIIXs should be investigated in otherwise healthy patients undergoing controlled and standardized hT. This opportunity could be offered by athyreotic patients diagnosed with differentiated thyroid cancer (DTC) that require radioiodine (RAI) treatment after total thyroidectomy. These subjects are often in excellent performance status (i.e., ECOG 0). Therefore, they usually undergo controlled, standardized hT by withdrawing levothyroxine (LT4) before RAI administration.

The aim of this study was to investigate the inflammatory pattern expressed according to BIIXs in a population which underwent standardized, controlled hT for four weeks before RAI. Individual determinations were also considered.

## 2. Methods

### 2.1. Institutional Thyroid Cancer Patients’ Management

Garibaldi Nesima Hospital is a high-volume tertiary care center in Catania (Italy). Patients diagnosed with DTC in the central and southeastern area of Sicily after thyroidectomy are usually placed on LT4 and then referred to the Thyroid Center of this hospital for subsequent treatments and long-term follow-up. According to the institutional guidelines, DTC patients who are candidates for RAI administration receive treatment within six months from total thyroidectomy. Patients assessed as otherwise healthy are prepared for RAI by a standard protocol of hT induction: (1) LT4 therapy is stopped four weeks before the day scheduled for RAI administration; and (2) during the first and second week, the patients assume l-triiodothyronine at a fixed daily dose, while they do not assume thyroid hormones during the third and fourth week. On the day of RAI administration, the patient is hospitalized, individual anthropometric parameters are recorded, and laboratory tests are assessed, including a hemogram and TSH, FT4, FT3, thyroglobulin (Tg), and anti-thyroglobulin antibody (AbTg) assessments, at the institutional laboratory department.

### 2.2. Case Selection

The records of DTC patients treated with RAI at our institution between January 2016 and December 2019 were screened. These included patients with the following criteria: adult, underwent total thyroidectomy, and RAI for the first time within six months after surgery. The exclusion criteria were the following: (1) age < 18 years; (2) concomitant therapy with corticosteroids or drugs with a known effect on hypothalamic pituitary thyrotropic activity (e.g., benzodiazepines, antidepressants, and antipsychotics) or anticoagulation drugs; (3) TSH below the target (i.e., ≥30 µU/mL) just before RAI; and (4) concomitant diagnosis potentially interfering with BIIX values (i.e., chronic inflammatory disease, autoimmune disease, acute or chronic infection, hematologic disease, heart failure, atrial fibrillation, myeloproliferative disorders, hepatic or renal disorders, and other endocrine or metabolic disorders including diabetes mellitus).

### 2.3. Data Extraction

The following data were extracted: the results of blood tests collected just before RAI (i.e., TSH, FT3, FT4, Tg, AbTg, creatinine, glycaemia, and complete blood count) and individual demographic and anthropometric parameters.

### 2.4. Measures

Among blood tests, beyond counting data, we calculated the following indexes: the neutrophils/lymphocytes ratio (NLR); platelets/lymphocytes ratio (PLR); monocytes/lymphocytes ratio (MLR); (neutrophils × platelets)/lymphocytes ratio (SII); (neutrophils × monocytes)/lymphocytes ratio (SIRI); and (neutrophils × platelets × monocytes)/lymphocytes ratio (AISI). The BMI and Tg were analyzed as categorical variables for a better clinical evaluation (i.e., BMI: healthy weight, overweight, or obesity; Tg: <10 or ≥10 ug/L). 

### 2.5. Statistical Analysis

The results were expressed as median values and an interquartile range (IQR). A Mann–Whitney U test was performed to compare the differences between two independent samples. The Kruskal–Wallis test was used to compare three or more independent groups. The correlations between variables were estimated using Pearson’s or Spearman’s correlation as appropriate. A *p* value < 0.05 was considered statistically significant. Statistical analyses were performed with Jamovi computer software (version 2.3).

## 3. Results

According to the above selection criteria, 143 patients were enrolled in the study. The sample included 100 females and 43 males with a median age of 43 (33–54) years. Among patients with available weights, 41 (40.2%) had normal weights, 31 (30.4%) were overweight, and 30 (29.4%) were obese. Seventeen patients assumed LT4 before total thyroidectomy at a lower dose than that used afterward. Table 1 illustrates the baseline characteristics of the study sample, including blood count data.

The median and IQR of the BIIXs were the following: an NLR of 2.09 (1.58–2.70); PLR of 111.43 (96.52–142.67); MLR of 0.19 (0.14–0.24); SII of 480 (374–640.91); SIRI of 0.83 (0.5–1.18); and AISI of 182.03 (118.2–290). Each BIIX showed a rather narrow IQR (Figure 1a–f).

The Mann–Whitney U test showed that males had significantly higher levels of white blood cells, neutrophils, and monocytes and lower platelets than females. There were no significant differences in lymphocytes, eosinophils, or basophils (Appendix A). Furthermore, the NLR (*p* < 0.001), MLR (*p* < 0.001), SIRI (*p* <  0.001), and AISI (*p* = 0.005) were significantly higher in males than females (Appendix A) (Figure 2a–d). The Kruskal–Wallis test showed statistical significance for the BMI categories of NLR (*p* = 0.034), SII (*p* = 0.015), SIRI (*p* = 0.003), and AISI (*p* = 0.005) levels (Appendix A) (Figure 3a–d). No significant correlations were found between the BIIXs and age, TSH, FT4, and FT3 (Appendix A). When blood data were explored according to Tg level, it was found that the patients with Tg > 10 ug/L had significantly higher NLR (*p* = 0.005), PLR (*p* = 0.04), MLR (*p* < 0.001), and SIRI (*p* = 0.02) values than those having Tg < 10 ug/L (Appendix A) (Figure 4a–d).

A sub-analysis was performed considering non-obese subjects. This subgroup consisted of 72 patients, 48 females, and 24 males, with a median age of 44 (33, 54.3) years. The median and IQR of the BIIXs were the following: NLR of 1.88 (1.46–2.57); PLR of 111 (91.6–141); MLR of 0.17 (0.14–0.25); SII of 442 (320–581); SIRI of 0.72 (0.49–1.03); and AISI of 173 (102–253). Therefore, each BIIX showed an even narrower IQR with respect to those of the general population. Table 2 shows the median values with the interquartile range for each BIIX calculated from the total sample and the sample of patients with known BMIs who were not obese, divided into males and females. The NLR (*p* = 0.02), MLR (*p* = 0.002), SIRI (*p* < 0.001), and AISI (*p* = 0.01) were significantly higher in males than females. No significant differences or correlations were found in the BIIX values based on age, TSH, FT4, or FT3. According to the Tg value, it was found that the patients with Tg ≥ 10 ug/L had significantly higher NLR (*p* = 0.002), MLR (*p* = 0.006), and SIRI (*p* = 0.04) levels than those having Tg < 10 ug/L, with no significant difference regarding the other BIIXs.

## 4. Discussion

Inflammation is a crucial process with multiple potential implications. Athyreotic patients undergoing controlled, standardized hT provide us with the great opportunity to explore the inflammation pattern associated with hT. This is the case for patients with DTC who require RAI and then undergo a predefined period without thyroid hormone replacement. All these cases are exposed to the same hT condition, and, in the absence of other comorbidities that affect the inflammation indexes, constitute a homogeneous population. Accordingly, we selected DTC patients who were otherwise fit and healthy and were managed with withdrawal of LT4 for a 4-week period and the complete absence of thyroid hormones assumed for a 2-week period before RAI. Our population included all subclasses of adulthood with a median age of 43 years, and the BMI classes were distributed almost equally. Furthermore, the female/male ratio reflected the usual demographic data reported for thyroid cancer. These findings indirectly confirm the high reliability of the study sample [29]. Therefore, the aim of this study was to perform a complete assessment of the inflammatory profiles in subjects with clinical or overt hT using BIIXs. The results merit careful attention.

Due to the lack of a control group, a qualitative analysis of the data would be appropriate for a full understanding of the achieved figures. In this way, the first evidence of this study was the narrow IQR of each BIIX (Figure 1a–f). This aspect allowed us to confidently describe the BIIX values of this select hT population which were influenced only by the anthropometric features. The analyses then showed a significant difference between males and females in terms of the NLR, MLR, SIRI, and AISI. Considering that the BIIXs are ratios, these differences cannot be explained by only the physiological difference in blood count between males and females. Rather, males and females could suffer differently from hT. In the first-level blood cell assessment, the males had higher numbers of white blood cells, neutrophils, and monocytes and a lower platelet number than the females, while there were no differences in the lymphocyte, eosinophil, and basophil numbers. Therefore, these consequent differences in the BIIXs could indicate a lesser bone marrow impairment in hematopoiesis in males exposed to hT. A compensatory role for testosterone might be advocated as a possible explanation of these data. Otherwise, a greater inflammatory response may occur in men. These same results regarding the NLR, MLR, SIRI, and AISI were confirmed even when only non-obese patients were considered. This denotes that during hT, sex plays a greater role than adipose mass in the inflammatory response. It has been clearly shown that sex has an important influence on the innate immune system. This is indeed due to gender-specific genetic differences. In addition, sex hormones alter the environmental milieu to which immune cells are exposed. The effects of sexual dimorphism on immune cells occur through classical nuclear hormone receptor signaling, non-classical hormone signaling, and downstream epigenetic remodeling [30]. In general, the physiological levels of estradiol enhance the pro-inflammatory capacity of the human immune system, and testosterone seems to play an overall anti-inflammatory effect, which may contribute to the dampened immune response in males [30]. Among the various mechanisms reported, estrogen has a strong influence on NF-kB signaling, which plays a key role in a variety of inflammatory and autoimmune processes [31], and testosterone suppresses 5-lipoxygenase activity and its production of leukotrienes, powerful vasoconstrictors, and chemoattractants [32]. Our results appear to be contrary to this evidence. However, the differences between males and females are quite remarkable. Therefore, hypothyroidism, the main interferent of this population, seems to have a greater pro-inflammatory effect in men than in women.

Second, age was proven to not be correlated with an effect on either the blood count or BIIXs. On one hand, this means that the inflammation response does not vary according to age. On the other hand, these data corroborate our case selection strategy and achieve a reliable sample without individual interfering factors.

Third, the analysis of the correlation between thyroid tests and BIIXs deserves a full discussion. TSH variation with hT induction was not correlated with the BIIXs. This finding was unexpected and leads to important considerations. In a condition of overt hT, the inflammatory response is probably lead by the duration of exposure to thyroid hormones lowering or being absent rather than the TSH levels [33]. One may wonder if the production of FT4 and FT3 by remnants of the thyroid could interfere with the observation of this relationship. However, these hormones also did not show any correlation with the BIIXs.

Therefore, TSH cannot assess hT severity alone. In patients with skewed thyroid tests, and especially in borderline conditions such as subclinical hT, inflammatory assessment could be useful to fully understand the real systemic impact of TSH increasing case by case.

Although TSH variation may reflect the presence of thyroid remnants or cancer persistence, data on the true thyroid-specific marker Tg were different from those of TSH. The NLR, MLR and SIRI were mostly higher in patients with a Tg ≥ 10 ug/L, with the most significative evidence being for the MLR. Since monocytes can infiltrate damaged tissue to remove it through the phagocytosis product of metastasis [34], we could speculate that these BIIXs, in particular the MLR, could have a role in predicting the long-term outcomes of DTC.

Finally, the NLR, SII, SIRI, and AISI were higher in obese patients, confirming the reliability of these BIIXs to assess the inflammatory status and the synergic action led by adiposity and hypothyroidism to trigger systemic inflammation [35]. 

To date, in the literature, there is much evidence on the role of hypothyroidism in erythropoiesis and consequent anemia [20,21]. However, studies on the effect of hypothyroidism on leukocytes and inflammation are scant, being conducted on small samples and with controversial results. Furthermore, they are primarily focused on chemotaxis rather than BIIXs. Only one article presents an investigation similar to our study, but its evaluation was limited to the NLR and PLR. It did not observe significant differences between the phases of euthyroidism, overt hypothyroidism, and subclinical hyperthyroidism in patients with DTC. That study was conducted on approximately 50 patients, did not take into account gender-specific differences, age, or BMI in the analysis of the BIIX levels, and did not indicate the period after thyroidectomy within which hypothyroidism was induced for RAI [36].

This study shows a complete inflammatory assessment in patients with overt hT by BIIXs. It provides reference values for the BIIXs for hT in the highly selective population. Moreover, the differences which emerged by analyses pose a first step to functional use of these indices in clinical practice.

The main limitation of this study is the absence of a control group, which may partially limit the discussion of the results. It would be useful to consider as potential control groups other clinical models or time-dependent monitoring, including thyroid disease-free patients, thyroidectomized patients with good thyroid hormonal compensation, and DTC patients with good prognoses. The resulting evidence would allow us to further define the values of the BIIXs associated with altered thyroid hormonal compensation (e.g., in thyroiditis or athyreotic patients) or correlated with a poor prognosis in DTC. Since the identified markers currently cannot inform clinical decisions, further prospective studies are needed to clarify these aspects.

Beyond the limitations of this study, it is necessary to note that the study population was highly selective without any important interfering factors. In fact, these patients were assessed as healthy, and thus they were addressed to RAI using an hT induction protocol based on withdrawal replacement therapy for 4 weeks rather than rhTSH (Thyrogen). All this has allowed an accurate analysis focused only on the inflammatory pattern of overt hT.

## 5. Conclusions

To the best of our knowledge, this is the first study to perform an inflammatory assessment via BIIXs in patients with controlled and standardized clinical or overt hT who were otherwise fit and healthy. It shows the features determining the differences in BIIX levels and a series of BIIX values to be useful as potential references for evaluating the inflammatory status in patients with TSH increases. Furthermore, the results suggest a potential role for the NLR, SII, SIRI, and AISI in assessing the inflammatory status in obese hypothyroid patients and for the NLR, MLR, and SIRI as inflammatory and possible prognostic markers in DTC patients.

## Figures and Tables

**Figure 1 biomedicines-12-00239-f001:**
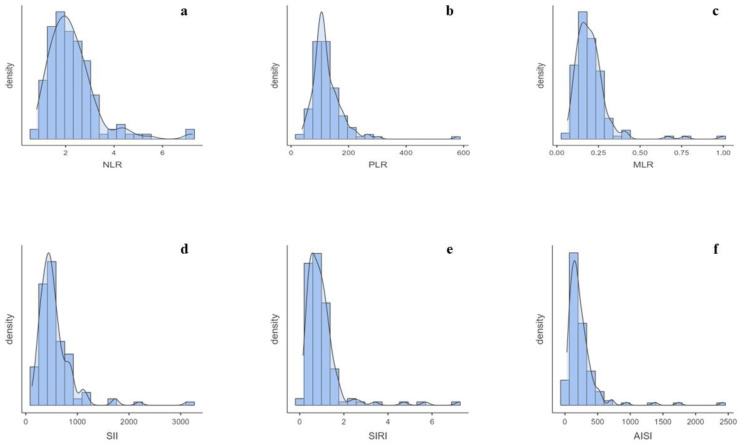
(**a**) Distribution of NLR data. (**b**) Distribution of PLR data. (**c**) Distribution of MLR data. (**d**) Distribution of SII data. (**e**) Distribution of SIRI data. (**f**) Distribution of AISI data.

**Figure 2 biomedicines-12-00239-f002:**
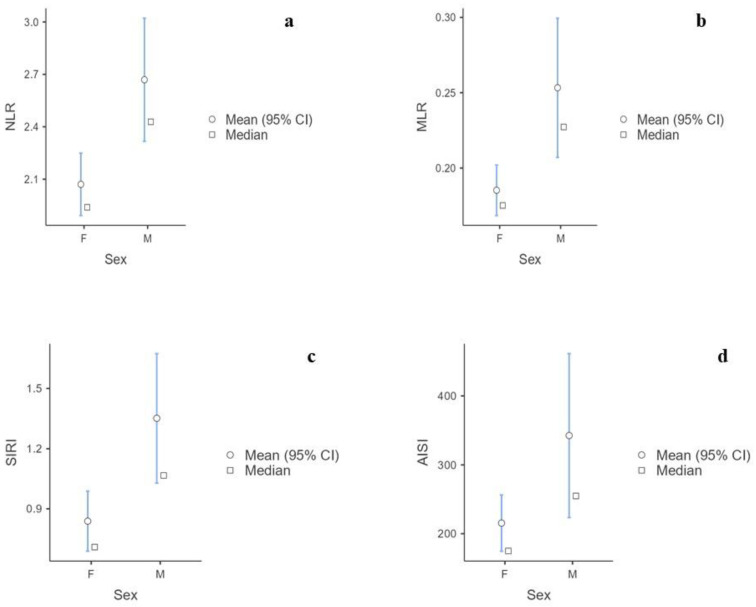
(**a**) Difference in NLR values according to sex. (**b**) Difference in MLR values according to sex. (**c**) Difference in SIRI values according to sex. (**d**) Difference in AISI values according to sex. F = females; M = males.

**Figure 3 biomedicines-12-00239-f003:**
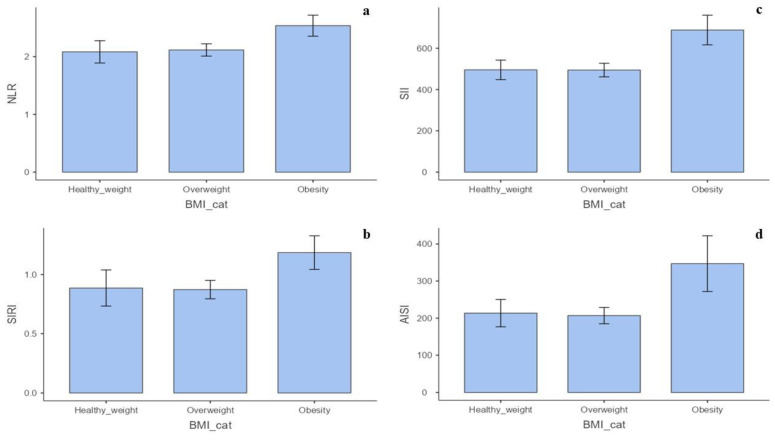
(**a**) Difference in NLR values according to BMI. (**b**) Difference in SII values according to BMI. (**c**) Difference in SIRI values according to BMI. (**d**) Difference in AISI values according to BMI.

**Figure 4 biomedicines-12-00239-f004:**
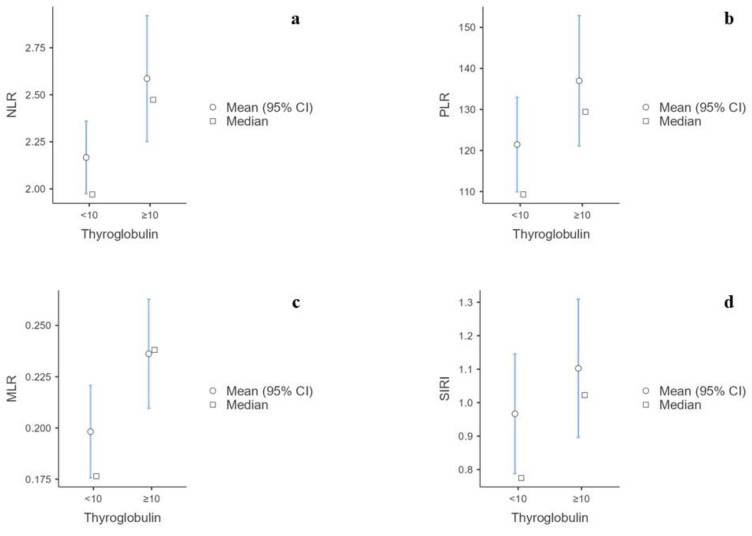
(**a**) Difference in NLR values according to thyroglobulin level. (**b**) Difference in PLR values according to thyroglobulin level. (**c**) Difference in MLR values according to thyroglobulin level. (**d**) Difference in SIRI values according to thyroglobulin level.

**Table 1 biomedicines-12-00239-t001:** Baseline biochemical and blood count features of the study sample. RBC = red blood cells; WBC = white blood cells.

			Percentiles	
	N	Median	25th	75th
RBC	143	5.00	4.67	5.40
Hemoglobin	143	13.90	13	14.90
WBC	141	7.20	5.80	8.20
Platelets	143	237	199.50	281
Neutrophils	141	4.30	3.50	5.20
Lymphocytes	141	2.00	1.70	2.60
Monocytes	141	0.40	0.30	0.50
Eosinophils	141	0.20	0.10	0.30
Basophils	141	0.10	0.00	0.10
Creatinine	142	0.93	0.83	1.08
Glycemia	141	86	78	96
Thyroglobulin	143	1.76	0.44	6.00

**Table 2 biomedicines-12-00239-t002:** Overview of median value and IQR limits in total sample and in non-obese population. TS = total sample; NOS = non-obese population.

				Percentiles			
	Sex	Median		25th		75th	
		TS	NOS	TS	NOS	TS	NOS
NLR	F	1.94	1.79	1.50	1.42	2.42	2.39
	M	2.43	2.38	1.90	1.83	3.15	2.88
PLR	F	115.44	111.96	97.62	95.45	146.41	148.29
	M	105.56	107.85	91.19	88.38	141.44	130.10
MLR	F	0.18	0.15	0.133	0.13	0.21	0.21
	M	0.23	0.23	0.17	0.17	0.26	0.26
SII	F	469.35	433.11	333.57	296.02	591.32	552.11
	M	512.43	487.55	393.24	386.48	758.15	716.83
SIRI	F	0.71	0.59	0.46	0.43	1.04	0.83
	M	1.07	0.95	0.81	0.72	1.37	1.35
AISI	F	174.89	150.70	106.57	91.37	261.62	200.83
	M	254.84	215.05	156.77	156.63	349.17	317.51

## Data Availability

The data sets used or analyzed during the current study are available from the corresponding author upon reasonable request.

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
