# Peer review of "Inflammatory Profile Assessment in a Highly Selected Athyreotic Population Undergoing Controlled and Standardized Hypothyroidism"

_biomedicines, 2024, doi:10.3390/biomedicines12010239_

Round 1
Reviewer 1 Report
Comments and Suggestions for Authors
The manuscript is a retrospective study that explores the inflammatory pattern expressed according to blood inflammatory indexes in a population that had undergone controlled hypothyroidism for four weeks before radioiodine. The topic is interesting and well-introduced. The methodology is described in detail. After reading this work, I have some observations:
1. I recommend rereading the text for typos and grammatical mistakes.
2. Avoid using abbreviations in the abstract and conclusions.
3. Figure 3 is not readable.
4. Line 270: add the institutional approval number.
Comments on the Quality of English LanguageThere are some typos and grammatical errors.
Author Response
Thank you very much for your precious suggestions. We have assimilated them in the article files, with consequent paper quality and clarity improvement. 1) We carefully reviewed the entire manuscript, correcting any typos and grammatical errors. 2) We tried to reorganize the abstract avoiding the acronyms, however it was not possible to report the main results and to respect the journal's rules for authors about the extension of abstract. Furthermore, acronyms such as NLR, PLR, SIRI and SII are now commonly used in medicine papers even in the absence of full definitions. 3) The figure has been modified by increasing the size of each graph with its descriptions, maintaining the font used in the descriptions of the other images. 4) It was added (Please see line 302).
Reviewer 2 Report
Comments and Suggestions for Authors
The study titled "Inflammatory profile assessment of a highly selected thyroidectomized population undergone controlled and standardized hypothyroidism" explores the inflammatory profile associated with hypothyroidism (hT) in athyreotic patients, specifically those with differentiated thyroid cancer (DTC). The investigation employs a well-defined population undergoing thyroid hormone withdrawal, providing valuable insights into the inflammatory relationships. The study effectively justifies the selection of athyreotic DTC patients undergoing thyroid hormone withdrawal, ensuring a homogenous population. The demographics, including age, BMI distribution, and gender ratio, align with typical characteristics observed in thyroid cancer populations. I appreciate the statistical analyses pragmatic and easy to understand. The investigation successfully analyzes sex-based differences in inflammation indices, providing insights into potential gender-specific responses to hypothyroidism. The consideration of obese/non-obese patients is useful and adds depth to the findings. Areas for improvement have been correctly discussed by the authors including the absence of a control group limiting the ability to draw definitive conclusions and the retrospective nature. A qualitative analysis, as suggested by the authors, would be essential for a more comprehensive understanding.
I have just minor suggestions:
To enhance transparency, consider incorporating a reflection on the absence of time-dependent monitoring of BIIXs.
Augmenting the discussion by suggesting potential sources for control data, even if not incorporated, would contribute to a more comprehensive examination.
A more in-depth exploration of the underlying physiological mechanisms driving gender differences could further enrich the discussion.
Additionally, it would be advantageous to elaborate on how the identified markers may inform clinical decisions.
Author Response
Thank you very much for your precious suggestions. We have assimilated them in the article, with consequent paper quality and clarity improvement. Please see lines 213-227; 267-275.
Reviewer 3 Report
Comments and Suggestions for Authors
Dear authors,
I found your manuscript very interesting and after a minor revision it can be published. Hypothyroidism is a disease including heterogeneous clinical symptoms and findings, and often it is not diagnosed. I found it very useful to study indices that can objectify systemic inflammation in patients with controlled and standardized clinical hT. You demonstrated that the features determining differences in BIIXs levels and a series of BIIXs values are useful as potential reference for evaluating inflammatory status in patients with TSH increase.
In the Discussion section, you should highlight better the results obtained, by comparison with other studies published on this topic. Also, the limitations of the study should be grouped at the end of the Discussion chapter (for example, lines 197-198 should be moved at the end).
Author Response
Thank you very much for your precious suggestions. We have assimilated them in the article, with consequent paper quality and clarity improvement. Please see lines 197; 256-262, 267-275.